# Wnt/β-catenin regulates an ancient signaling network during zebrafish scale development

**Andrew J Aman, Alexis N Fulbright[†], David M Parichy\***

Department of Biology and Department of Cell Biology, University of Virginia, Charlottesville, United States

**Abstract** Understanding how patterning influences cell behaviors to generate three dimensional morphologies is a central goal of developmental biology. Additionally, comparing these regulatory mechanisms among morphologically diverse tissues allows for rigorous testing of evolutionary hypotheses. Zebrafish skin is endowed with a coat of precisely patterned bony scales. We use *in-toto* live imaging during scale development and manipulations of cell signaling activity to elucidate core features of scale patterning and morphogenesis. These analyses show that scale development requires the concerted activity of Wnt/β-catenin, Ectodysplasin (Eda) and Fibroblast growth factor (Fgf) signaling. This regulatory module coordinates Hedgehog (HH) dependent collective cell migration during epidermal invagination, a cell behavior not previously implicated in skin appendage morphogenesis. Our analyses demonstrate the utility of zebrafish scale development as a tractable system in which to elucidate mechanisms of developmental patterning and morphogenesis, and suggest a single, ancient origin of skin appendage patterning mechanisms in vertebrates.

DOI: https://doi.org/10.7554/eLife.37001.001

**\*For correspondence:**
dparichy@virginia.edu

**Present address:** [†]Huntsman Cancer Institute, University of Utah, Salt Lake City, United States

**Competing interests:** The authors declare that no competing interests exist.

## Introduction

Developmental patterning generates distinct gene expression regimes that regulate morphogenetic cell behaviors. Identifying core regulatory modules, elucidating the specific interactions they comprise, and how these activities are translated into discrete morphological outcomes are central goals of modern developmental biology. To these ends, considerable progress has been made at embryonic stages, yet patterning and morphogenesis at post-embryonic stages remain poorly understood.

Skin appendages are classic developmental model systems that have been leveraged to generate insights into how mesenchymal–epithelial signaling interactions pattern tissues and affect morphogenesis (*Lai and Chuong, 2016*). These structures, including the hair, teeth, mammary and eccrine glands of mammals, feathers and scales of birds, and scales or scutes of reptiles are among the most conspicuous features of the adult form. In addition to being developmental biology models with clear biomedical relevance, skin appendages have been of longstanding interest for their evolutionary significance. Although amniote skin appendages such as hairs and feathers have widely varied morphologies, recent work suggests that all of these appendages derive from a common progenitor that was present in stem amniotes (*Di-Poï and Milinkovitch, 2016*; *Wu et al., 2018*).

Another skin appendage is the scale of fishes. In many extant teleosts, scales comprise thin, overlapping plates of dentin-like calcified extracellular matrix embedded in the dermis (*Sire et al., 2009*). Similar to amniote skin appendages, fish scales develop relatively late in ontogeny and are distributed across the skin in a tight, hexagonal grid pattern in the adult. Scales have been the object of excellent histological and ultrastructural studies that have characterized developmental anatomy, and genetic analyses that have identified phenotypes associated with Ectosyslasin-A (Eda)

**eLife digest**   Hair, feathers or scales cover the skin of most land animals. Despite their apparent diversity, these appendages share many features: they are mainly formed of the protein keratin, are produced by the topmost layer of the skin and they start to form with skin cells moving inwards to form a pit. Across species, the same genes are also involved in controlling the development of these structures. This suggests that they have all evolved from a shared ancestral appendage, which may have been fish scales. However, scales in fish are formed of bones, not keratin, and they come from a different skin layer.

Here, Aman et al. explore the molecular mechanisms that control how zebrafish scales form and get their shape, which is a little-studied area of research. Cells at the surface of the fish were imaged live on the animal as they were developing and creating scales. The experiments involved manipulating the genetic information of these cells to tease out the molecular mechanisms that drive the creation of scales. This revealed that the genes that control the formation of the scales and of the appendages of land animals are the same and interact in similar ways. In particular, scales also require the skin to form a pit to develop, and the same genes direct this process in zebrafish and in furred or feathered creatures.

The work by Aman et al. suggests that all skin appendages, regardless of being sported by fish, birds or mammals, descend from the same structure. It also puts forward the zebrafish and its scales as a good model for scientists to study so they can understand better how certain hair and teeth disorders arise in humans.

DOI: https://doi.org/10.7554/eLife.37001.002

and Fgf mutations (*Harris et al., 2008*; *Rohner et al., 2009*; *Sire et al., 1997*). Nevertheless, these serially reiterated, highly accessible organs have yet to be exploited as a model for understanding skin appendage development at a cellular level. There is also uncertainty as to whether fish scales and amniote skin appendages are homologous, that is, derived from a single archetype organ in a common ancestor (*Paul, 1972*; *Sharpe, 2001*). If all vertebrate skin appendages are homologous, we would expect that common developmental regulatory mechanisms underlie teleost scales, avian feathers, mammalian hair and other appendage types.

Here, we use conditional-genetic manipulations, live imaging and gene expression assays optimized for post-embryonic fish to show that multiple signaling pathways, including Wnt/β-catenin, Eda, Fgf and Shh regulate scale patterning and morphogenesis. These analyses show that scale development relies on signaling interactions similar to interactions that regulate the patterning and morphogenesis of amniote skin appendages such as hair and feathers, and support a model in which diverse skin appendages of vertebrates arose from a common archetype. Additionally, we uncover a novel process of HH-dependent collective cell migration that is necessary for epidermal invagination during skin appendage morphogenesis. Together, our analyses establish the fundamental parameters that govern scale initiation and morphogenesis and lay the groundwork for exploiting zebrafish scale development as a system in which to discern general principles of developmental patterning, regulation of morphogenetic cell behaviors, and the evolution of genetic regulatory mechanisms.

## Results

### sp7+ osteoblast like cells generate scales and reveal amniote-like skin patterning in zebrafish

Histological, ultrastructural and fate mapping studies showed that scale morphogenesis begins with the formation of a bi-layered dermal papilla immediately beneath the epidermis (*Mongera and Nüsslein-Volhard, 2013*; *Shimada et al., 2013*; *Sire et al., 1997*). Yet these studies did not unambiguously identify the cell types involved. We hypothesized that, due to their calcified composition, scales will be formed by osteoblast-like cells. To test this, we analyzed the distribution of osteoblasts using fish transgenic for a previously described reporter with osteoblast-specific expression, *sp7: EGFP* (*DeLaurier et al., 2010*). *sp7* (formerly *osterix*) encodes a zinc finger transcription factor that is necessary and sufficient for osteoblast differentiation from committed progenitors (*Zhang, 2012*).

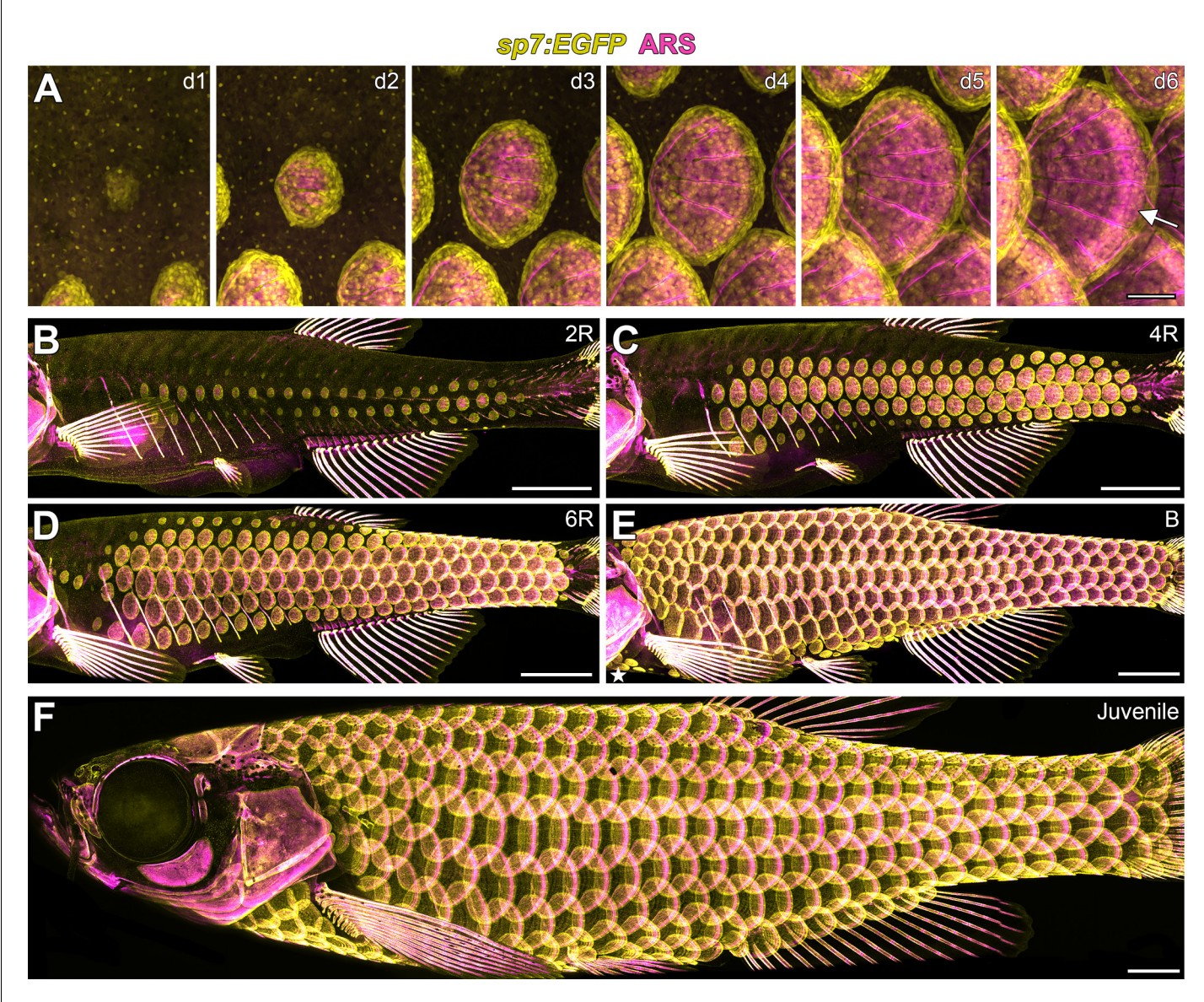

**Figure 1.** Zebrafish scale development. All images from a single representative individual labelled with *sp7:EGFP* (yellow, to visualize osteoblast-like scale forming cells) incubated with Alizarin Red-S vital dye (ARS, magenta, to visualize calcified matrix). (**A**) Development of a single, representative scale. A small cluster of dSFCs (d1) polarizes and grows posteriorly (d2–d5) until it imbricates with its neighbors and forms an intensely calcified limiting layer (arrow in d6). (**B–E**) Scales are added sequentially across the skin of larval fish. (**B**) Initially, two rows of scales form near the horizontal myoseptum, defined here as the 2R (two row) stage. (**C**) Additional dorsal and ventral rows arise, defining the 4R (four row) stage. (**D**) Row addition continues while columns of scales appear anterior and posterior, leading to the 6R (six row) stage. (**E**) In the B (beard) stage, a cluster of scales arises ventral–posterior to the head (star) and spreads along the ventrum. (**F**) Final squamation pattern in a juvenile. Scale bars, 200 μm (**A**); 1 mm (**B–F**).

DOI: https://doi.org/10.7554/eLife.37001.003

The following figure supplements are available for figure 1:

**Figure supplement 1.** Scales are deposited by *sp7:EGFP*+ osteoblast like cells.

DOI: https://doi.org/10.7554/eLife.37001.004

**Figure supplement 2.** Squamation details revealed by repeated live imaging of individual fish.

DOI: https://doi.org/10.7554/eLife.37001.005

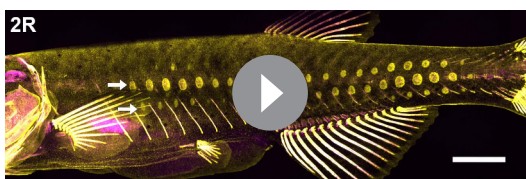

**Video 1.** Zebrafish squamation stages *sp7:EGFP* cells (yellow) and Alizarin Red-S labelled calcified matrix (magenta) serially imaged in live fish. Two row (2R), four for (4R), six row (6R) and beard (B) stages are annotated. Scale bar = 1 mm.
DOI: https://doi.org/10.7554/eLife.37001.006

Simultaneously, we monitored the distribution of calcified extracellular matrix using Alizarin Red S (ARS) vital dye (*Adkins, 1965*; *DeLaurier et al., 2010*).

Weak *sp7:EGFP* expression was first detected in clusters of cells in the skin, immediately followed by detection of calcified matrix (*Figure 1A* d1–d2 and *Figure 1—figure supplement 1A and B*). Throughout subsequent scale development, the distribution of calcified matrix correlated with the distribution of *sp7:EGFP+* cells (*Figure 1A*, d2–d6; *Figure 1—figure supplement 1B*) suggesting that *sp7:EGFP* labeled the dermal cell population that deposits calcified extracellular matrix comprising the scale plate. Hereafter, we refer to this cell population as dermal Scale Forming Cells (dSFCs). Early *sp7:EGFP+* dSFC clusters that lacked detectable calcified matrix showed a bi-layer papilla corresponding with structures described in histological and ultrastructural studies (*Figure 1—figure supplement 1C*). Therefore, *sp7:EGFP* labels developing scales from very early in morphogenesis and likely earlier than other scale osteoblast markers (*Iwasaki et al., 2018*).

Following their initial appearance, scales grew centripetally from the scale focus with a posterior bias leading to polarized extension through the addition of dSFCs and calcified matrix (*Figure 1—figure supplement 1D,E*). In fully formed scales, *sp7:EGFP+* dSFCs were arranged in a monolayer along the deep aspect of the calcified matrix but also looped around to cover the superficial surface, overlapping with the intensely ARS-labeled limiting layer (*Figure 1A*, *Figure 1—figure supplement 1F–H*).

Skin appendage primordia in amniotes spread sequentially across the skin during development (*Chuong et al., 2013*; *Dalle Nogare and Chitnis, 2017*; *Painter et al., 2012*; *Sick et al., 2006*). *In-toto* repeated live imaging of individual zebrafish revealed similar sequential addition of scale primordia (*Figure 1B–1F*, *Video 1*). The first scales were found in a row on the caudal peduncle followed by a second row that formed on the flank above the ribs (*Figure 1—figure supplement 2A, B*). Within one day, two complete rows of scales formed (*Figure 1B*). Additional rows were added dorsally and ventrally, and columns were added anteriorly and posteriorly (*Figure 1B–1D*; *Figure 1—figure supplement 2F–I*). Finally, a third scale origin appeared just posterior to the mandible and spread along the ventral surface to meet the ventral row scales just posterior to the pectoral fin insertion (*Figure 1E*; *Figure 1—figure supplement 2J*). These events lead to a complete coat of scales arranged in a half-offset, hexagonal grid on the juvenile fish (*Figure 1F*). Under optimized conditions, this process took ~13 days (*Figure 1—figure supplement 2C–E*). Identical scale patterning can be visualized using sequential calcium vital dye labelling (*Figure 1—figure supplement 2K–O*).

There are presently no staging conventions for squamation in zebrafish. Based on our live imaging, we propose a staging system for this process. Two rows (2R) represents the initial appearance of scales up to two complete scale rows. Four rows (4R) are fish with four complete scale rows, one dorsal and one ventral to the original two scale rows. This stage corresponds with the stage of grossly apparent posterior squamation (SP) defined in (*Parichy et al., 2009*). Finally, beard (B) stage fish have scales along the dorsal anterior, as well as the initiation of post-mandibular scales, corresponding with the anterior squamation (SA) stage defined in (*Parichy et al., 2009*).

Our observations of scale development indicate that sequential addition of skin appendage primordia, leading to a tightly packed hexagonal grid, is a feature of epidermal appendage development common to both amniotes and anamniote fishes. In turn, this suggests the hypothesis that these diverse appendage types are patterned by common mechanisms. To evaluate this hypothesis we next tested requirements for specific signaling activities during scale development.

## Wnt/β-catenin signaling is necessary for scale initiation

If amniote skin appendages and teleost scales arose from a common ancestral organ, we predicted that Wnt/β-catenin signaling should be necessary for scale development as it is for ectodermal

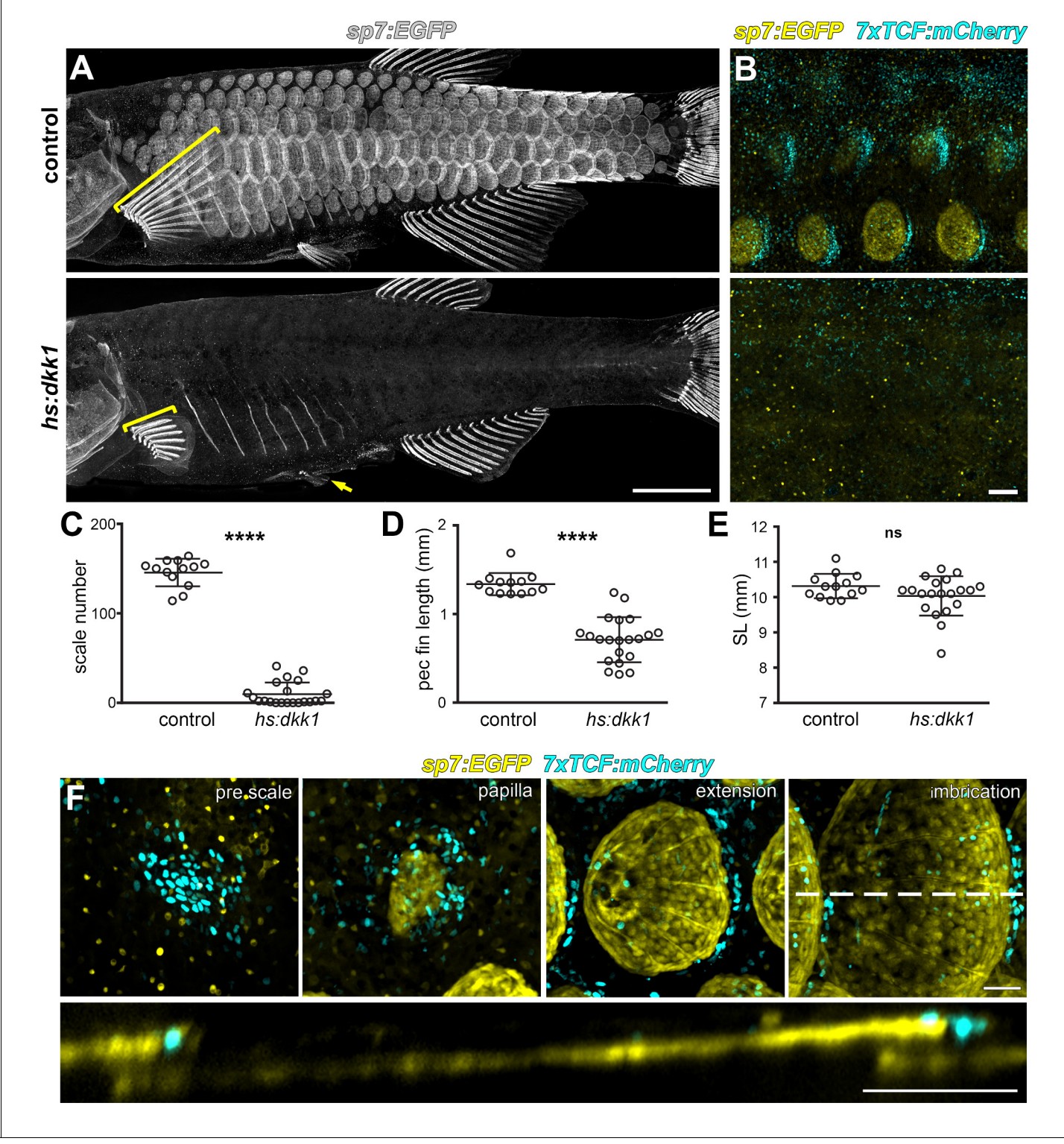

**Figure 2.** Wnt/β-catenin signaling is necessary for scale development. (**A**) Wnt repression by heat-shock induction of *hs:dkk1* transgene blocked scale formation, arrested fin outgrowth (yellow brackets) and prevented pelvic fin ray development (yellow arrow). (**B**) *hs:dkk1* induction abrogated patterned expression of the Wnt/β-catenin reporter transgene *7x:TCF:mCherry* (cyan). (**C**) *hs:dkk1* larvae developed significantly fewer scales (n = 21) then heat-shocked controls (n = 13; p<0.0001). (**D**) Pectoral fins were significantly shorter in *hs:dkk1* larvae (n = 21) than in heat-shocked controls (n = 13; p<0.0001). (**E**) *hs:dkk1* larvae achieved a similar standard length (n = 21) to heat-shocked controls (n = 13; p=0.1). Plots indicate means ± SD. (**F**) Live image series of Wnt/β-catenin activity revealed by expression of *7xTCF:mCherry* reporter transgene (cyan). Wnt/β-catenin reporter expression was first
*Figure 2 continued on next page*

*Figure 2 continued*

detected in a patch of epidermal cells in prospective scale regions (pre scale). Subsequently, *sp7:EGFP*+ dSFCs appeared underneath the *7xTCF: mCherry* expressing cells (papilla). During later scale extension and imbrication, *7xTCF:mCherry* expression persisted at the posterior margin in cells superficial and posterior to the scale forming cells. Scale bars, 1 mm (**A**); 100 μm (**B,F**).

DOI: https://doi.org/10.7554/eLife.37001.007

The following figure supplement is available for figure 2:

**Figure supplement 1.** Distribution of Wnt/β-catenin signaling targets in developing scales.

DOI: https://doi.org/10.7554/eLife.37001.008

appendages of amniotes (*Andl et al., 2002*; *Dhouailly et al., 2017*). To inhibit Wnt/β-catenin signaling during scale development, we used the *hs:dkk1* transgenic line that allows conditional expression of a potent and selective Wnt/β-catenin signaling inhibitor (*Glinka et al., 1998*; *Stoick-Cooper et al., 2007*). As predicted, inhibiting Wnt/β-catenin beginning prior to the appearance of *sp7:EGFP*+ papillae prevented scale formation (*Figure 2A and C*). This early treatment also blocked localized expression of the Wnt/β-catenin signaling activity reporter *7xTCF:mCherry* (*Figure 2B*) (*Moro et al., 2012*).

Beyond scale phenotypes, Wnt/β-catenin inhibition prevented fin outgrowth and formation of pelvic fin rays (*Figure 2A and D*), demonstrating a role for this pathway in fin development, in addition to previously documented functions in fin regeneration (*Kawakami et al., 2006*; *Stoick-Cooper et al., 2007*; *Wehner et al., 2014*). Scale and fin phenotypes were not caused by a generalized retardation of development, as Dkk1-overexpressing fish achieved sizes similar to those of non-transgenic, heat-shocked sibling controls (*Figure 2E*).

If Wnt/β-catenin signaling is necessary for initiating scale development, signaling activity must precede overt scale morphogenesis. Live imaging of fish doubly transgenic for *7xTCF:mCherry* and *sp7:EGFP* showed that, as predicted, Wnt/β-catenin signaling activity was evident before *sp7:EGFP* + scale papilla had formed (*Figure 2F*). After papilla formation, Wnt/β-catenin signaling became polarized toward the posterior scale margin, where it remained throughout scale growth. Because scales develop in an invariant sequence we were also able to analyze expression of conserved Wnt/β-catenin targets *lef1* and *axin2* in fixed specimens (*Cadigan and Waterman, 2012*; *Hovanes et al., 2001*; *Jho et al., 2002*; *Ramakrishnan and Cadigan, 2017*). In agreement with the *7xTCF:mCherry* reporter, *lef1* and *axin2*, as well as nuclear localized β-catenin, were found in skin patches presaging the appearance of scales (*Figure 2—figure supplement 1A,B*, pre scale and papilla; *Figure 2—figure supplement 1E,F*). Expression subsequently became restricted to a posteriorly biased ring around the circumference of the developing scale (*Figure 2—figure supplement 1A,B*, extension; *Figure 2—figure supplement 1E,G,H*). Although expression dynamics of these Wnt/β-catenin activity markers were broadly similar, *axin2* was expressed exclusively in dermal cells and was absent from the epidermis, whereas *lef1* was expressed in both dermis and epidermis (*Figure 2—figure supplement 1A,B*). As expected for Wnt target genes, early Dkk1 induction prevented patterned expression *lef1* and *axin2* (*Figure 2—figure supplement 1C,D*, early *hs:dkk1*). However, Dkk1-mediated Wnt repression initiated after scales had formed attenuated expression of *axin2*, but not *lef1* (*Figure 2—figure supplement 1C,D*, late *hs:dkk1*), suggesting a mechanism independent of Wnt/β-catenin signaling for maintaining *lef1* expression during scale outgrowth.

Taken together, expression of Wnt/β-catenin target genes prior to papilla appearance and lack of scales in Dkk1 overexpressing fish demonstrates that Wnt/β-catenin signaling is necessary for initiation of scale development. We next sought to address how Wnt/β-catenin signaling interacts with other signaling pathways during scale development.

## Zebrafish scale development requires interactions between Wnt/β-catenin and Eda signaling

The phenotypes of fish inhibited for Wnt/β-catenin signaling—abrogated scale and fin formation — were similar to phenotypes of fish lacking Ectodysplasin-A (Eda) signaling (*Harris et al., 2008*; *Kondo et al., 2001*). We therefore hypothesized that Wnt/β-catenin and Eda signaling interact during scale development. To elucidate regulatory linkages between these pathways we used the zebrafish *nkt* mutant line, harboring a presumptive *eda* loss of function allele (*Harris et al., 2008*). *nkt*

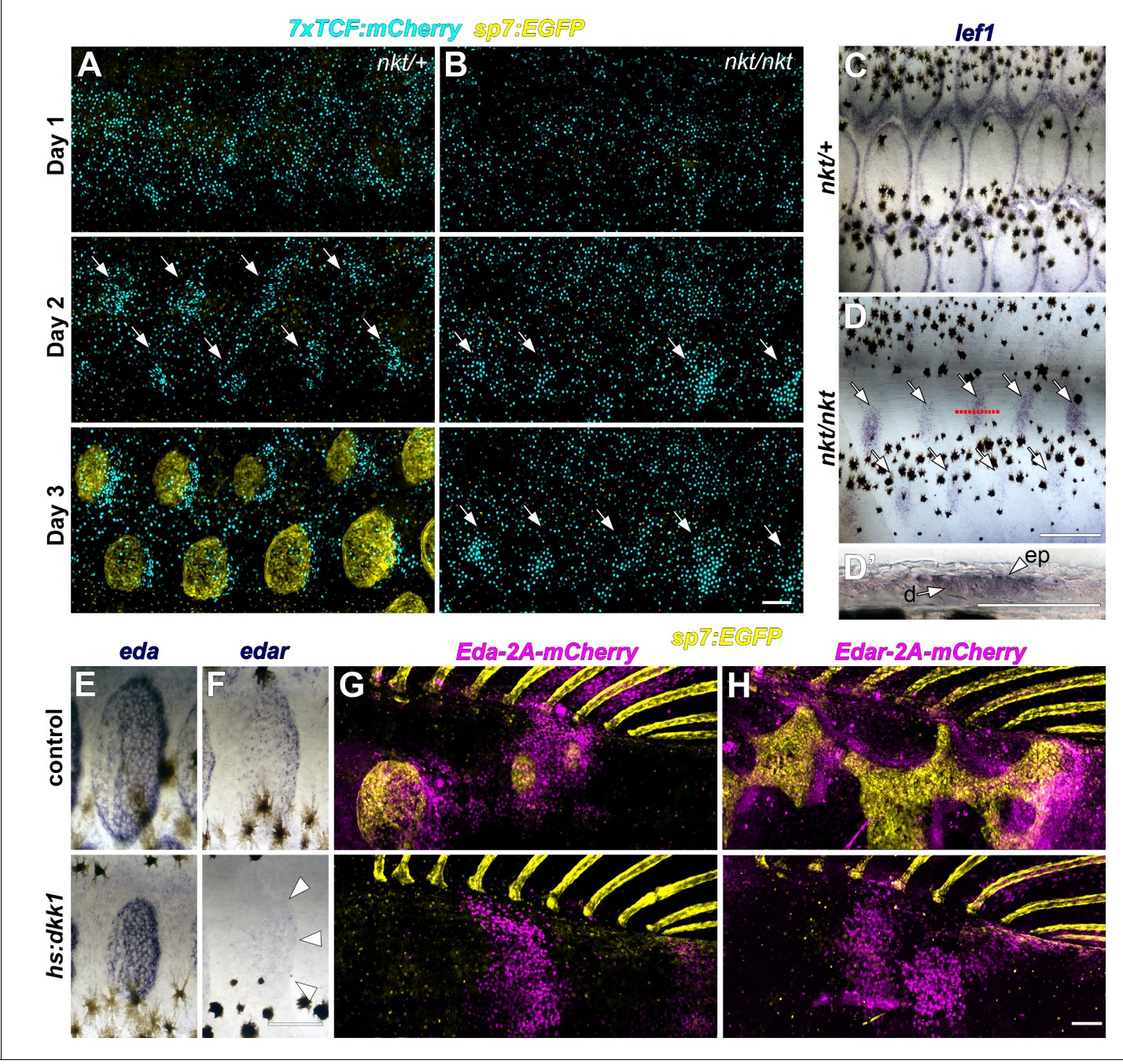

**Figure 3.** Wnt/β-catenin and Eda signaling interact during scale development. (A) In WT (*nkt/+*) larvae, Wnt/β-catenin activity revealed by expression of *7x:TCF:mCherry* (cyan), was patterned into a series of half-offset spots (arrows) prior to appearance of *sp7:EGFP*+ dSFCs (yellow, n = 6). (B) In Eda mutant (*nkt/nkt*) larvae, *7x:TCF:mCherry* expression was patterned into spots (arrows), although no dSFCs developed (n = 6). (C) WT (*nkt/+*) fish fixed at 10.0–10.5 mm had multiple rows of *lef1*+ scales (n = 6). (D,D') Scale-less Eda mutant (*nkt/nkt*) larvae of similar size had patterned expression of *lef1* (arrows) in basal epidermal (ep) and dermal (d) cells (n = 4 of 6). (E) *hs:dkk1* had no effect on *eda* expression in scales that developed prior to induction. (F) *hs:dkk1* attenuated *edar* expression in scales (arrowheads indicate position of scale margin). (G,H) Expression of ectopic Eda and Edar (both magenta) caused ectopic scale formation in heat-shocked controls (top: Eda, n = 10; Edar, n = 20) but not *hs:dkk1* larvae (bottom: Eda, n = 16; Edar n = 18). Scale bars, 100 μm (A, E–H); 50 μm (C, D, D').

DOI: https://doi.org/10.7554/eLife.37001.009

The following figure supplements are available for figure 3:

**Figure supplement 1.** Eda signaling during zebrafish scale development.

DOI: https://doi.org/10.7554/eLife.37001.010

*Figure 3 continued on next page*

*Figure 3 continued*

**Figure supplement 2.** Only epidermis is competent to respond to Eda signaling.
DOI: https://doi.org/10.7554/eLife.37001.011

mutants lacked overt indications of scale morphogenesis and *sp7:EGFP+* dSFCs in the skin, in addition to having impaired fin outgrowth (*Figure 3—figure supplement 1A*). Yet, live imaging of *sp7:EGFP; 7xTCF:mCherry* transgene expression and in situ hybridization against *lef1* showed that Wnt/β-catenin activity was patterned into spots in the epidermis of scale-less *nkt* mutants (*Figure 3A–3D*). Wnt/β-catenin targets were initially expressed in scale-appropriate patterns in *nkt* mutants, yet further rows were not formed and expression domains did not polarize. These findings indicate that initiation and patterning of Wnt activity in the skin is Eda-independent, whereas later refinement and reiteration of Wnt activity requires Eda signaling. The absence of overt scale development despite initiation of Wnt signaling, also demonstrates that Wnt signaling alone is not sufficient to induce scales in the absence of Eda.

Conversely, we asked whether Eda signaling requires Wnt/β-catenin activity. We first examined the expression of genes encoding Eda and its receptor Edar during normal scale development. Immediately prior to scale formation, *eda* expression disappeared from epidermal cells above forming scale papillae where *edar* expression was first detected (*Figure 3—figure supplement 1B,C* pre-scale). Subsequently, *eda* was expressed broadly in dSFCs whereas *edar* expression became localized principally to the posterior margin of the scale epidermis (*Figure 3—figure supplement 1B,C* papilla, extension) in the vicinity of Wnt/β-catenin target genes (*Figure 2F* and *Figure 2—figure supplement 1*). These similar expression patterns (and common phenotypes of pathway blockade) suggested the hypothesis that *edar* is a Wnt/β-catenin signaling target during scale formation, whereas *eda* expression might be independent of Wnt/β-catenin. We therefore induced *hs:dkk1* following the appearance of initial scales (2R stage) and assayed expression of *eda* and *edar*. As predicted, *edar* expression was strongly attenuated in Dkk1-overexpressing, Wnt/β-catenin inhibited fish, whereas *eda* expression persisted (*Figure 3E,F*).

Lack of *edar* expression in Wnt/β-catenin inhibited fish, taken together with residual Wnt activity in *nkt* mutants that nevertheless formed no scales, suggested that a lack of Eda pathway signaling is the primary reason that scales do not develop in Wnt inhibited fish. To test this, we conditionally misexpressed Eda and Edar while manipulating Wnt/β-catenin signaling. In fish with normal Wnt/β-catenin signaling, misexpression of either Eda or Edar initiated scale development, as revealed by accumulations of *sp7:EGFP+* dSFCs in regions where scales do not normally form at these stages (*Figure 3G,H*, control). Misexpression of Eda in either epidermis or dermis initiated ectopic scale development (*Figure 3—figure supplement 2A,B*), whereas Edar did so only when misexpressed in the epidermis (*Figure 3—figure supplement 2C,D*), suggesting that competence to respond to Eda signaling is unique to epidermis. However, Eda/Edar driven ectopic scale induction was completely blocked in the absence of Wnt/β-catenin signaling (*Figure 3G–3H*, *hs:dkk1*). Therefore, neither Wnt/β-catenin nor Eda signaling is sufficient to trigger scale development in the absence of the other.

## Fgf-dependent differentiation of scale forming cells requires Wnt/β-catenin signaling

Fgf signaling has been implicated in skin appendage development of amniotes (*Huh et al., 2013*; *Mandler and Neubüser, 2004*; *Petiot et al., 2003*) and in scale development of teleosts: zebrafish harboring mutations simultaneously in *fgfr1a* and *fgf20a* have scales that are smaller than normal, whereas fish mutant for *fgfr1a* alone have scales that are larger than normal. It remains unclear whether Fgf signaling is required for scale initiation, outgrowth or both (*Daane et al., 2016*; *Rohner et al., 2009*). To circumvent potential functional redundancies and to test the necessity of Fgf signaling for scale development, we employed the pan-Fgf receptor inhibitor BGJ398 that has been shown to specifically block activity of Fgf receptor kinase but not other closely related kinases (*Guagnano et al., 2011*). This treatment led to an immediate and complete arrest of squamation and scale growth without affecting overall somatic growth of the fish (*Figure 4A,B*; *Figure 4—figure supplement 1A–C*), demonstrating that Fgf signaling is necessary for both scale initiation and outgrowth. To determine which Fgf receptors are involved in scale development, we generated cDNA

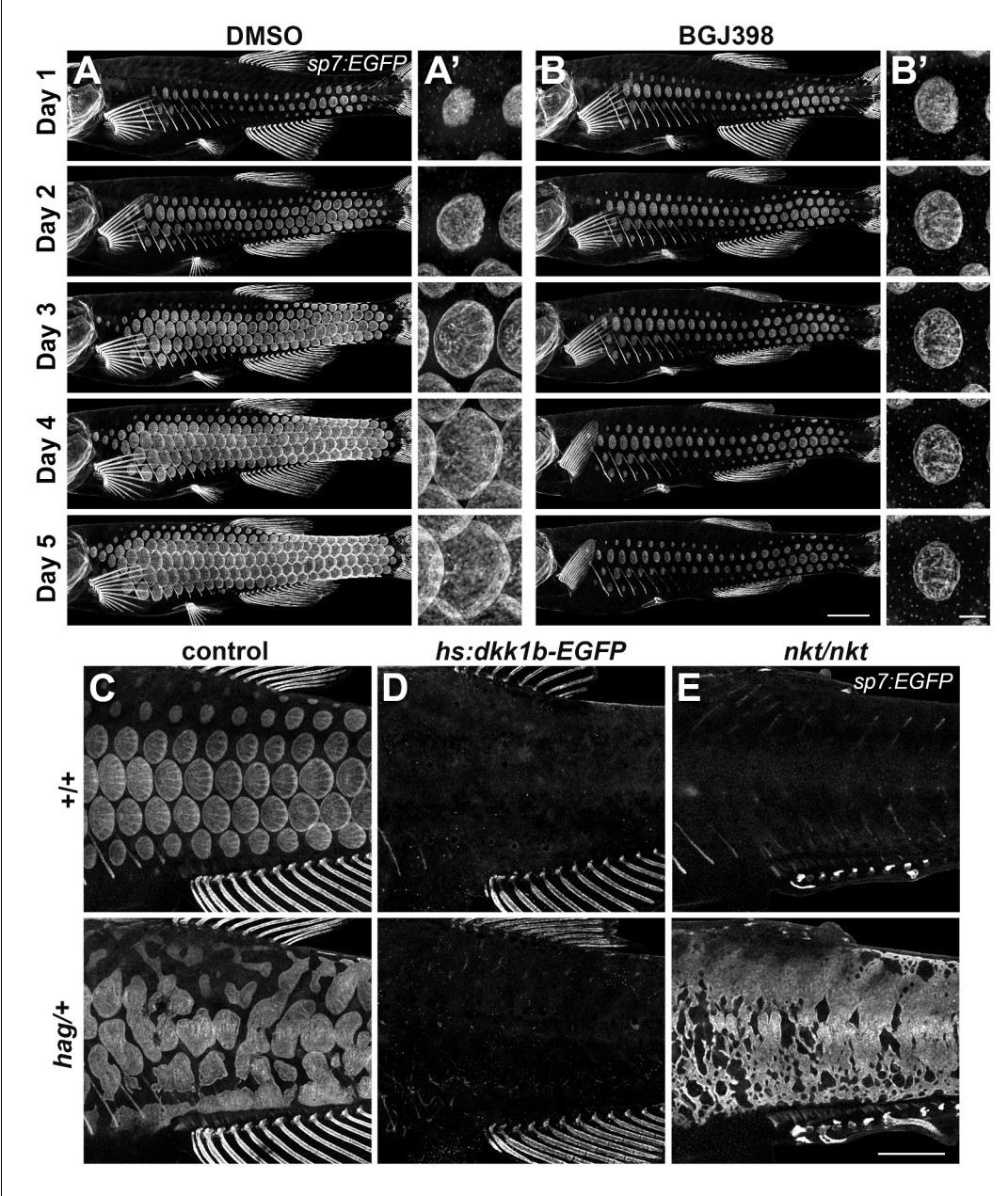

**Figure 4.** An Fgf–Wnt/β-catenin signaling interaction is necessary for differentiation of scale forming cells.  (A,B) Imaging *sp7:EGFP* over a five day treatment with 1 µM BGJ398 to inhibit Fgf receptors revealed completely arrested squamation (n = 8), whereas DMSO-treated control fish developed normally (n = 8). (A',B') BGJ398 treatment also arrested growth of individual scales. (C) *hag/+* mutants overexpress Fgf8a and had disorganized differentiation of *sp7:EGFP*+ dSFCs (n = 6). (D) dSFCs did not appear in *hag/+* mutant skin with *hs:dkk1* induction (n = 12). (E) Loss of Eda signaling in *nkt/nkt* mutants did not prevent differentiation of *sp7:EGFP*+ scale forming cells, likely owing to residual Wnt activity in the skin (n = 12). Scale bars, 1 mm (A–E); 100 µm (A', B').

DOI: https://doi.org/10.7554/eLife.37001.012

The following figure supplement is available for figure 4:

**Figure supplement 1.** Fgfr signaling blockade, expression and persistence after Wnt/β-catenin signaling inhibition.

DOI: https://doi.org/10.7554/eLife.37001.013

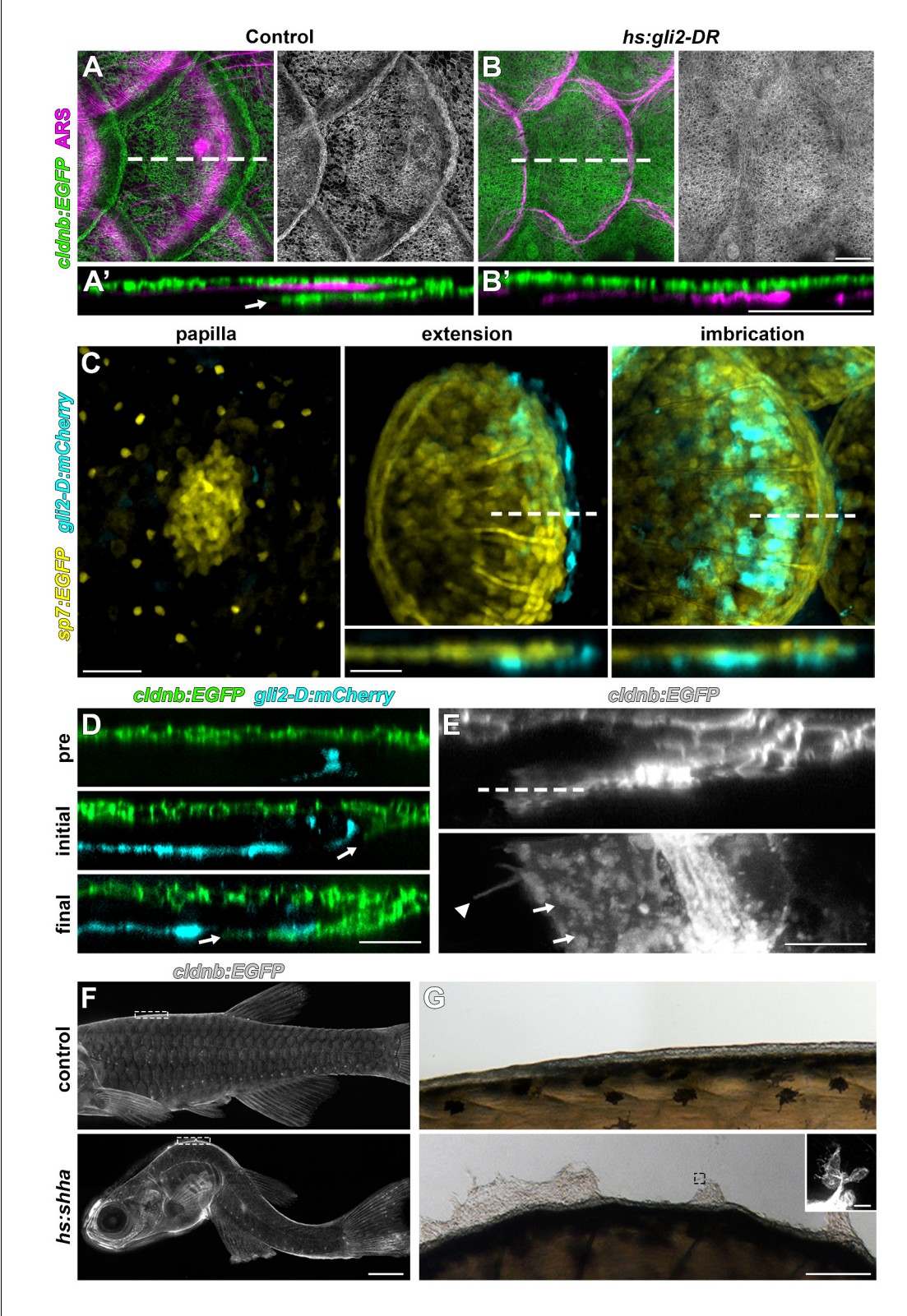

**Figure 5.** HH signaling is necessary for epidermal morphogenesis. (**A**) In heat-shocked control fish the epidermis (labeled by *cldnb:EGFP*, green) was folded around the posterior margin of individual scales (marked by ARS, magenta). (**A'**) orthogonal projection, arrow indicates leading edge of epidermis; n = 8. (**B**) Heatshock induction of a dominant repressor Gli2 (*hs:gli2-DR*), completely blocked epidermal folding. (**B'**) Orthogonal view; n = 12. (**C**) HH-responding cells (cyan) were very few at early stages of scale development (papilla) labelled by *sp7:EGFP* (yellow). HH-responding cells

*Figure 5 continued on next page*

*Figure 5 continued*
were found posterior and deep to the extending scale (extension) and persisted throughout later scale growth (imbrication); n = 8. (**D**) HH-responding cells (cyan) appeared prior to epidermal invagination (pre). Initial invagination (arrow), followed the contour of the HH responding population (initial). Epidermal invagination continued in close association with HH responding cells (final); n = 8. (**E**) High resolution imaging of the leading row of invaginating epidermal cells revealed hallmarks of cell migration including membrane ruffles (arrows) and long cellular extensions (arrow head); n = 10. (**F**) Shha overexpression by heatshock induction of *hs:shha* lead to severe kyphosis; n = 48. (**G**) *cldnb:EGFP*+ epidermis in *hs:shha* larvae migrated from the animal onto the glass coverslip during image acquisition; n = 6. Scale bars, 100 μm (**A,B**); 50 μm (**C**); 25 μm (**D**); 10 μm (**E**); 1 mm (**F**); 100 μm (**G**); 20 μm (**G**, inset).
DOI: https://doi.org/10.7554/eLife.37001.014
The following figure supplement is available for figure 5:

**Figure supplement 1.** HH signaling during scale development.
DOI: https://doi.org/10.7554/eLife.37001.015

from 4R stage skin and assayed by RT-PCR the expression of each of the five Fgf receptor genes present in the zebrafish genome. Of these, amplicons for *fgfr1a*, *fgfr1b* and *fgfr2* were recovered. While riboprobes against *fgfr1b* did not yield tissue specific staining, consistent with previous reports (*Rohner et al., 2009*), *fgfr1a* was detected in dSFCs and *fgfr2* in the epidermal posterior margin during scale development (*Figure 4—figure supplement 1D,F*).

To determine how Fgf signaling integrates with Wnt and Eda signaling during scale development, we used *hagoromo* (*hag*) mutants that overexpress Fgf8a in the skin post-embryonically due to a viral insertion upstream of *fgf8a* (*Amsterdam et al., 2009*; *Kawakami et al., 2000*). We found that *hag/+* fish develop large, disorganized sheets of *sp7:EGFP*+ dSFCs (*Figure 4C*), a previously over-looked phenotype. In *hag/+* fish with repressed Wnt/β-catenin signaling, neither sheets nor foci of *sp7:EGFP*+ dSFCs developed in response to *fgf8a* overexpression (*Figure 4D*). By contrast, *hag/+* fish simultaneously homozygous for *nkt*—and so lacking Eda but retaining residual Wnt signaling activity (*Figure 3A–3D*)—formed broad sheets of *sp7:EGFP*+ dSFCs (*Figure 4F*). Taken together these results show that Fgf-mediated differentiation of dSFCs requires Wnt/β-catenin, but not Eda signaling. The Wnt-dependence of Fgf signaling is not likely due to modulation of Fgfr gene expression (*Figure 4—figure supplement 1E,G*) but could reflect direct or indirect regulation of other Fgf pathway components yet to be identified.

## HH signaling requires Wnt/β-catenin and Eda pathways and is necessary for epidermal morphogenesis of scales

HH signaling is necessary for morphogenesis of amniote skin appendages (*Bitgood and McMahon, 1995*; *Dassule and McMahon, 1998*; *St-Jacques et al., 1998*), and *shha* transcript has been detected in developing zebrafish scales (*Harris et al., 2008*; *Iwasaki et al., 2018*; *Sire and Aki-menko, 2004*). To elucidate the role of HH signaling, we used a heat-shock inducible dominant repressor form of Gli2 (DR-Gli2) and treatment with the Smo antagonist cyclopamine (*Gould and Missailidis, 2011*; *Shen et al., 2013*). These treatments did not affect induction or patterning of scales, but did alter scale morphogenesis. In normal scale development, the epidermis folds around the growing posterior margin of the developing scale (*Figure 5A*). Strikingly, epidermal folding was completely blocked by HH repression (*Figure 5B*). Correspondence of DR-Gli2 and cyclopamine phenotypes (*Figure 5—figure supplement 1A*) and repression of HH pathway targets (*Figure 5—figure supplement 1B*) confirmed specificity and efficacy. These results also demonstrated that epidermal morphogenesis during scale development is an active, HH-dependent process and not simply a passive consequence of scale plate growth, indeed, epidermal folding was completely absent even when underlying scale plates overlapped (*Figure 5—figure supplement 1A*).

We next investigated the expression dynamics of HH ligand-encoding *shha*, the conserved HH transcriptional target *hhip*, and a transgenic reporter of HH signaling, *gli2-D:mCherry* (*Chuang and McMahon, 1999*; *Lum and Beachy, 2004*; *Schwend et al., 2010*). Since HH repression did not interfere with scale induction we predicted that expression of HH pathway genes and signaling activation would appear later in scale morphogenesis. We found that—unlike Wnt/β-catenin, Eda and Fgf signaling—HH signaling began only after papilla morphogenesis (*Figure 5C*). *shha* was initially expressed in the epidermis overlying the scale papilla and was later restricted to a column of two to three cells at the posterior scale margin (*Figure 5—figure supplement 1D*). *gli2-D:mcherry* and *hhip*

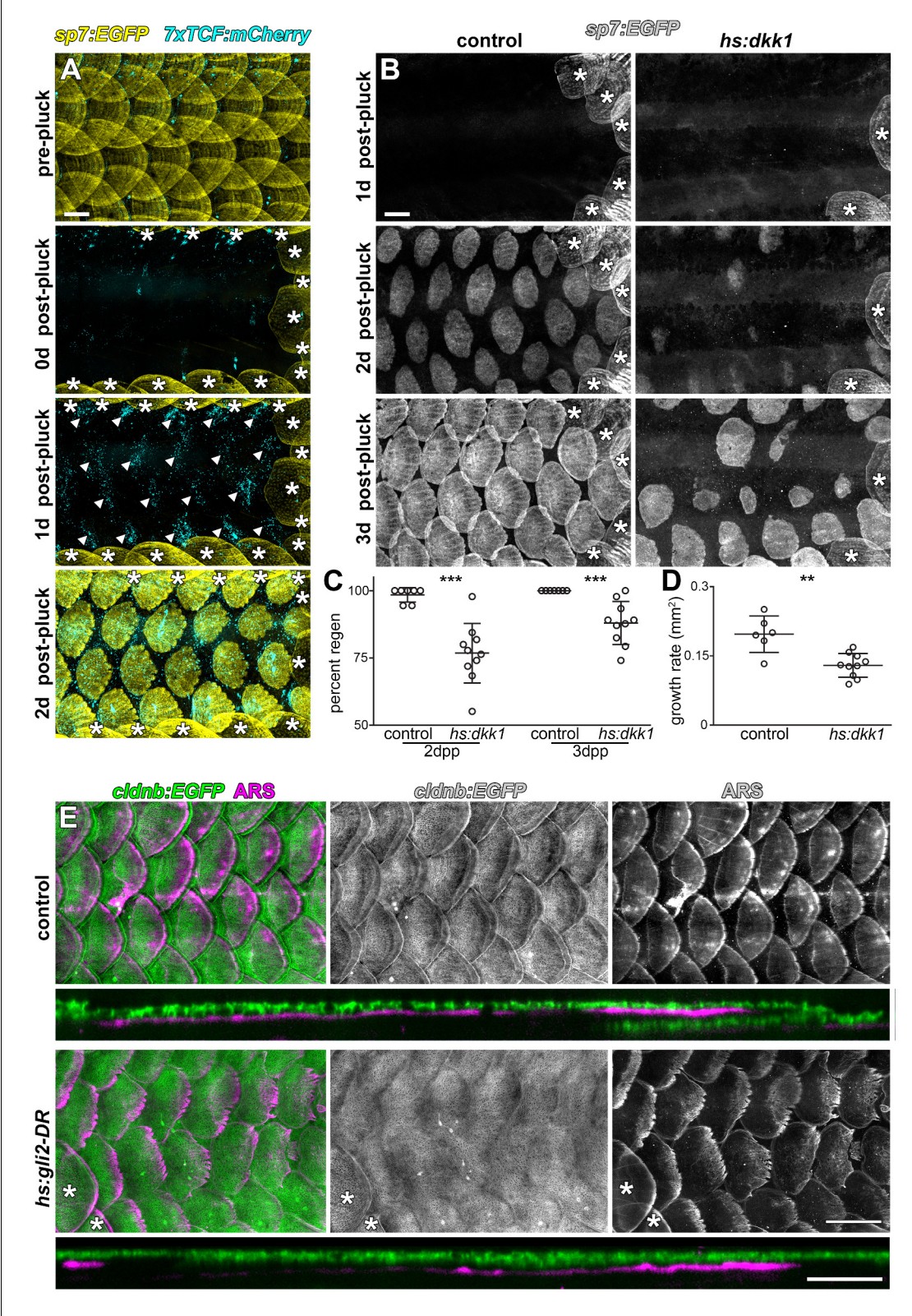

**Figure 6.** Wnt/β-catenin and HH signaling are necessary for normal scale regeneration. (A) Adult scales labelled with *sp7:EGFP* (yellow), were removed from the caudal peduncle (0 d post-pluck). One day following surgery (1 d post-pluck) patterned Wnt/β-catenin signaling activity was detected, as revealed by *7xTCF:mCherry* (cyan, arrowheads). After two days (2 d post-pluck), all removed scales had started to regenerate; n = 8. (B) Scale regeneration occurred normally in heat-shocked controls (n = 7), whereas *hs:dkk1* induction delayed scale regeneration (n = 10). Asterisks (*) mark

*Figure 6 continued on next page*

Figure 6 continued

ontogenic scales not plucked. (C) Heat-shocked controls regenerated a significantly higher proportion of scales two days after scale removal (2dpp) then *hs:dkk1* fish (p=0.0004). Three days after scale removal (three dpp), all heat-shocked controls had regenerated their full complement of scales whereas *hs:dkk1* fish regenerated significantly fewer scales (p=0.0007). (D) Regenerating scales grew more quickly in heat-shocked controls than in *hs:dkk1* fish (p=0.003). Plots indicate means ± SD. (E) In heat-shocked controls, epidermis (*cldnb:EGFP*, green) folded around regenerated scale plates (ARS; magenta) within ten days of scale removal (n = 6), whereas HH repression (*hs:gli2-DR*) prevented epidermal folding (n = 8). Asterisks (*) mark ontogenic scales not plucked. Scale bars, 500 μm (A, B, E); 100 μm (E, orthogonal views).

DOI: https://doi.org/10.7554/eLife.37001.016

were detected in a population of cells beneath the *sp7:EGFP+* dSFCs, where expression persisted throughout scale morphogenesis (*Figure 5C*; *Figure 5—figure supplement 1E*).

To further investigate the requirement for HH signaling in epidermal morphogenesis, we imaged fish doubly transgenic for the epidermal transgene *cldnb:EGFP* and *gli2-D:mCherry* throughout scale development. This revealed *cldnb:EGFP+* epidermal cells invaginating into the underlying dermis in close association with the HH-responding population (*Figure 5D*). Live imaging of cells at the leading edge of invaginating epidermis revealed hallmarks of active invasive migration including broad lamellipodia and long cellular extensions (*Figure 5E*).

If HH signaling triggers epidermal invagination by promoting invasion, we reasoned that Shha overexpression might promote excessive or ectopic invagination (*Shen et al., 2013*). Unfortunately, Shha overexpression in fish transgenic for *hs:shha-GFP* led to rapid kyphosis, failure to feed and developmental arrest, precluding analyses of scale morphogenesis (*Figure 5F*). Strikingly, however, epidermal cells of these fish rapidly moved from the surface of the fish to the glass coverslip (*Figure 5G*). Cells at the leading edge of this sheet were *cldnb:EGFP+* and displayed numerous long filopodia, consistent with active migration. Similar phenotypes have been observed in early larval skin and are associated with impaired epidermal cohesion and increased invasiveness (*Boggetti et al., 2012*; *Carney et al., 2007*).

Because expression of *shha* in the epidermal posterior margin overlaps with expression of Wnt/β-catenin target genes and with *edar* (*Figure 2F*, *Figure 2—figure supplement 1A,B,E–H*; *Figure 3—figure supplement 1C*), we hypothesized that *shha* expression might require Wnt/β-catenin signaling, Eda signaling, or both. We therefore assayed expression of HH signaling components while manipulating Wnt/β-catenin and Eda pathways. *shha*, *hhip* and *gli-D:mCherry* expression were strongly attenuated in both Wnt-repressed and *nkt* mutant fish (*Figure 5—figure supplement 1F–H*). Since *nkt* mutants retain epidermal Wnt signaling activity (*Figure 3B*) but lack Eda, these data suggest that *shha* expression is regulated by Eda signaling during scale development.

## Wnt/β-catenin and HH signaling are required for scale regeneration

During normal life, fish often lose scales owing to interactions with other species and the environment, and scales have long been recognized for their regenerative ability (*Smith, 1931*). To further test roles for early (Wnt/β-catenin) and late (HH) signaling pathways, and whether functions are conserved in both ontogenetic and regenerative contexts, we removed all scales from the caudal peduncle of adult transgenic fish. One day after scale removal, patterned *7xTCF:mCherry* expression revealed Wnt/β-catenin signaling in the epidermis, presaging the appearance of regenerated scales (*Figure 6A*). To test the requirement for this activity, we repressed Wnt/β-catenin signaling during scale regeneration by Dkk1 induction. This resulted in the regeneration of significantly fewer, slower growing scales as compared to controls (*Figure 6B–6D*). Finally, to test the role of HH signaling during regeneration we expressed DR-Gli2 following scale removal. As during scale ontogeny, HH-repression completely blocked epidermal morphogenesis (*Figure 6E*). Therefore, Wnt/β-catenin and HH signaling play similar roles during scale development and regeneration.

## Discussion

Our analysis, based on live imaging of individual fish, has revealed the developmental anatomy of squamation and individual scale development. We find that zebrafish squamation proceeds through an invariant sequence of row and column addition (*Figure 7A*), this differs from previous reports based on fixed specimens that inferred a sequential spread of scales from posterior to

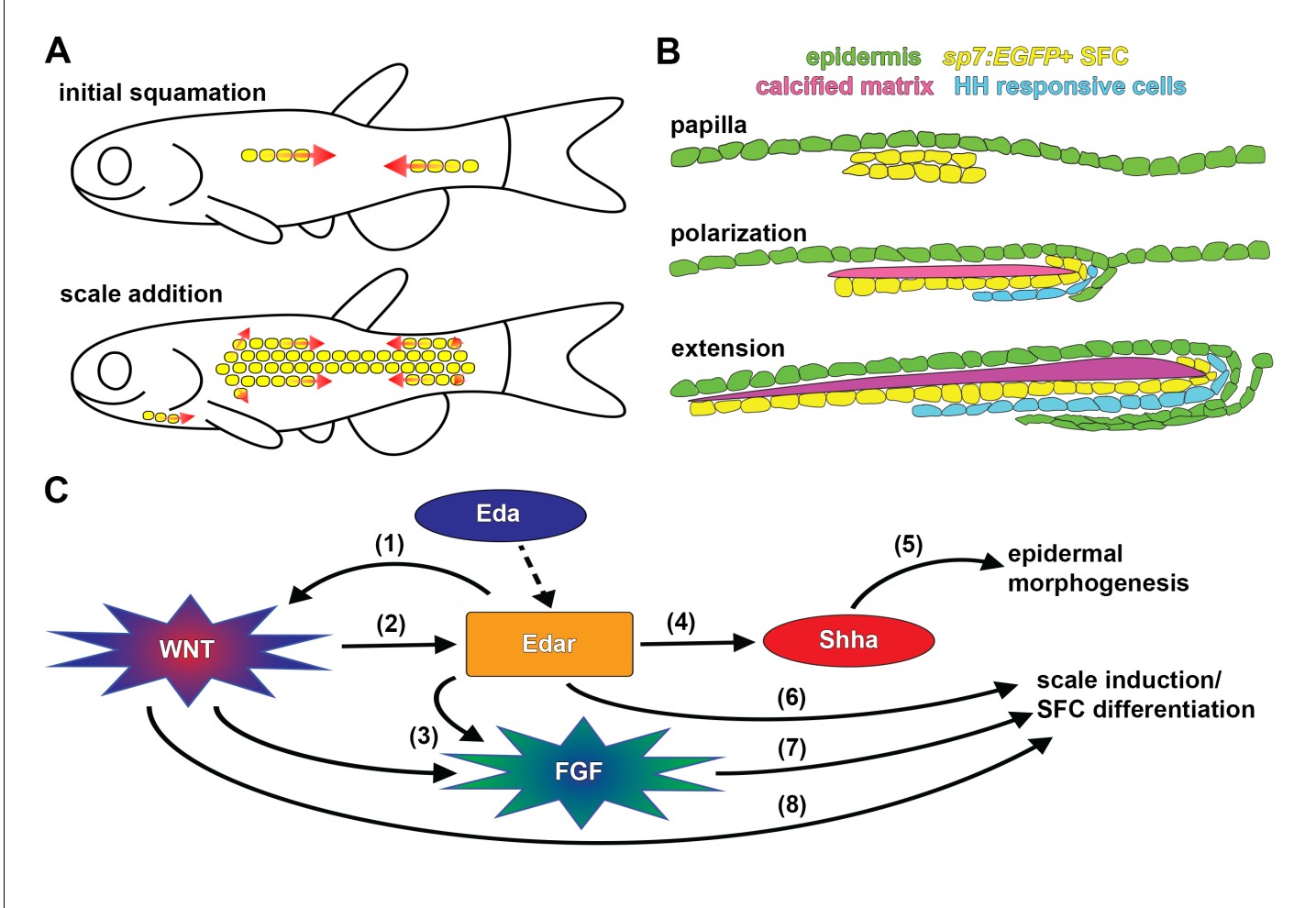

**Figure 7.** Summary of scale development and schematic of molecular interactions. (**A**) Zebrafish squamation sequence showing the addition of scales (yellow) by addition of rows and column (arrows). (**B**) Scale morphogenesis is first recognized as at the papilla stage when *sp7:EGFP+* dSFCs (yellow) differentiate beneath the epidermis (green). Following papilla formation polarized growth commences and scale forming cells line the posterior margin and deep aspect of the growing calcified plate (magenta). At this stage epidermal invagination begins. These processes (polarized growth of the calcified matrix and epidermal invagination) continue to generate the final morphology of the scale. (**C**) Schematic representation of molecular interactions leading to scale initiation and epidermal morphogenesis. (1) Refinement and reiteration of Wnt/β-catenin activity required Eda signaling (*Figure 3A–D*). (2) Wnt/β-catenin signaling was necessary for edar expression (*Figure 3F*). (3) Wnt/β-catenin and Eda signaling were necessary for patterned expression of Fgf receptors (*Figure 4—figure supplement 1E,G*). (4) Eda signaling was necessary for *shha* expression and HH pathway activity (*Figure 5—figure supplement 1F–H*). (5) HH signaling was necessary for epidermal morphogenesis (*Figure 5A,B*; *Figure 5—figure supplement 1A*). (6) Edar misexpression caused ectopic differentiation of dSFCs (*Figure 3H*, *Figure 3—figure supplement 2C*). (7) Broadly overexpressed Fgf8a caused broad, disorganized differentiation of dSFCs via a Wnt dependent mechanism (*Figure 4C–E*). (8) Neither Eda, nor Fgf signaling was sufficient to cause differentiation of dSFCs in the absence of Wnt signaling (*Figure 3G,H*; *Figure 4D*). Dashed line denotes activation of Edar by the Eda signaling ligand.

DOI: https://doi.org/10.7554/eLife.37001.017

anterior (*Sire et al., 1997*). The sequence we documented resembles that of goldfish (*Carassius auratus*) and medaka (*Oryzias latipes*)(*Iwamatsu, 2014*; *Li et al., 2015*). Since the lineages leading to zebrafish and medaka are thought to have diverged over 300 million years ago (*Near et al., 2012*), the squamation sequence presented here potentially represents a conserved, ancestral character of teleosts. Zebrafish squamation is also strikingly reminiscent of the ordered addition of chicken feather and reptile scale anlagen, suggesting this mode of patterning is conserved throughout vertebrates (*Di-Poï and Milinkovitch, 2016*; *Jung et al., 1998*).

Previous histological and ultrastructural studies have characterized the basic anatomy of scale development (*Lippitsch, 1990*; *Sire et al., 1997*; *Sire and Akimenko, 2004*). We infer that *sp7:*

*EGFP+* osteoblast like cells (dSFCs) deposit the calcified scale plate. Based on the morphology and distribution of these cells, they likely correspond to dermal cells identified in previous fate-mapping (*Lee et al., 2013*; *Mongera and Nüsslein-Volhard, 2013*; *Shimada et al., 2013*). Our analyses of dSFC distribution in live animals showed that these cells first appear as a bi-layered condensation, corresponding to the dermal papilla identified in histological analyses (*Sire et al., 1997*). Since dermal papilla formation represents the very first overt indication of scale morphogenesis, *sp7:EGFP* proved to be a useful marker for scale initiation. By labelling epidermis and using vital dyes that reveal calcified matrix, we were able to monitor key cell populations involved in scale formation. Combining these assays with visualization of signaling pathway activity revealed the distribution of signaling event during scale morphogenesis. For example, we find that a sheet of HH-responding cells develop beneath nascent scales and presumably coordinate scale extension (*Iwasaki et al., 2018*) with epidermal invagination (*Figure 7B*), through intermediary mechanisms that are currently not known.

Forward genetic screens have implicated Eda and Fgf signaling in scale development (*Harris et al., 2008*; *Kondo et al., 2001*; *Rohner et al., 2009*). The discovery that Eda signaling is necessary for development of both amniote skin appendages and fish scales suggested a single origin of skin appendages in ancient fishes (*Di-Poï and Milinkovitch, 2016*; *Harris et al., 2008*; *Houghton et al., 2005*; *Kondo et al., 2001*; *Zhang et al., 2009*). Yet, it remains debated whether involvement of Eda signaling implies homology between amniote and fish skin appendages or rather a general functional requirement of Eda for epithelial–mesenchymal signaling interactions (*Sharpe, 2001*). We found that Eda and Wnt/β-catenin signaling integrate similarly during fish scale and amniote skin appendage development: (i) Wnt/β-catenin signaling was initiated in the absence of Eda signaling yet pattern refinement and reiteration were Eda-dependent (*Figure 3A–D*); and (ii) Wnt/β-catenin signaling was necessary for the expression of the Eda receptor, *edar* (*Figure 3F*; *Houghton et al., 2005*; *Zhang et al., 2009*).

Simultaneous manipulation of multiple signaling activities revealed that while Eda and Fgf signaling are sufficient to drive ectopic differentiation of dSFCs and scale development, these pathways were able to do so only in the presence of functional Wnt/β-catenin signaling. Since dSFCs do not differentiate in Wnt-inhibited fish with simultaneous upregulation of Eda or Fgf, there are likely other, as yet unidentified, parallel mechanisms by which Wnt regulates scale development (*Figure 7C*). These interactions are again similar to signaling interactions previously described for hair and feather patterning in amniotes (*Andl et al., 2002*; *Houghton et al., 2005*; *Huh et al., 2013*; *Liu et al., 2008*; *Mandler and Neubüser, 2004*; *Petiot et al., 2003*; *Sick et al., 2006*).

During late steps in scale development, we found that Wnt/β-catenin– Eda dependent HH signaling regulates epidermal morphogenesis. Since HH signaling is also necessary for epidermal morphogenesis during hair and feather development (*McKinnell et al., 2004*; *St-Jacques et al., 1998*), it is possible that some functional outputs of skin appendage signaling networks are conserved and anciently evolved. In turn, this suggests a previously unappreciated role for invasive migration in epidermal morphogenesis that may be of general relevance to understanding cellular mechanisms underlying skin appendage development and regeneration (*Armstrong et al., 2017*). It will be interesting to learn how modifications to these terminal processes have contributed to diversity in skin appendage morphologies across vertebrates.

Taken together, our analyses of teleost scale developmental genetics using zebrafish are consistent with a single origin of skin patterning mechanisms in ancient fishes that has been conserved in extant vertebrates, even as the final adult morphology of feathers, hairs and scales appear wildly divergent. Importantly, the fossil record indicates that early tetrapods were endowed with fish-like calcified dermal scales (*Coates, 1996*; *Jarvik, 1996*), with a progressive loss of calcified matrix over geological time (*Mondéjar-Fernández et al., 2014*). In light of conserved developmental regulatory architecture, this suggests a scenario in which skin appendages lost dermal calcified matrix and gained epidermal keratinization ultimately leading to the skin appendages of extant amniotes. Conservation of molecular mechanisms that regulate skin appendage patterning and early morphogenesis enables the use of zebrafish scale development as a model system for understanding vertebrate skin patterning and morphogenesis with exceptional opportunities for live imaging and forward genetic analysis, complementing existing chicken and mouse models.

# Materials and methods

## Key resources table

| Reagent type (species) or resource | Designation | Source or reference | Identifiers | Additional information |
|---|---|---|---|---|
| Gene (*Danio rerio*) | *fgfr1a* | This paper | NCBI_Reference_Sequence: NM_152962.3 | Amplified from cDNA |
| Gene (*Danio rerio*) | *fgfr2* | This paper | NCBI_Reference_Sequence: NM_001243004.1 | Amplified from cDNA |
| Gene (*Danio rerio*) | *shha* | This paper | NCBI_Reference_Sequence: NM_131063.3 | Amplified from cDNA |
| Gene (*Danio rerio*) | *hhip* | This paper | NCBI_Reference_Sequence: NM_001080012.1 | Amplified from cDNA |
| Gene (*Danio rerio*) | *lef1* | This paper | NCBI_Reference_Sequence: NM_131426.1 | Amplified from cDNA |
| Gene (*Danio rerio*) | *axin2* | This paper | NCBI_Reference_Sequence: NM_131561.1 | Amplified from cDNA |
| Gene (*Danio rerio*) | *eda* | This paper | NCBI_Reference_Sequence: NM_001115065.1 | Amplified from cDNA |
| Gene (*Danio rerio*) | *edar* | This paper | NCBI_Reference_Sequence: NM_001115064.2 | Amplified from cDNA |
| Strain, strain background (*Danio rerio*) | gli2-D:mCherry; Tg(Mmu.Foxa2-cryaa :mCherry)$^{nu15Tg}$ | Gift. PMID:21203590 | RRID:ZDB-ALT-110310-6 | NA |
| Strain, strain background (*Danio rerio*) | hs:gli2-DR; Tg(hsp70l:gli2aDR-EGFP)$^{umz33Tg}$ | Gift.PMID:23441066 | RRID:ZDB-TGCONSTRCT -130123-11 | NA |
| Strain, strain background (*Danio rerio*) | hs:shha; Tg(hsp70l:shha-EGFP)$^{umz30Tg}$ | Gift.PMID:23441066 | RRID:ZDB-ALT-130123-8 | NA |
| Strain, strain background (*Danio rerio*) | cldnb:EGFP; Tg(−8.0cldnb:LY-EGFP)$^{zf106Tg}$ | Gift.PMID:16678780 | RRID:ZDB-TGCONSTRCT -070117-15 | NA |
| Strain, strain background (*Danio rerio*) | WT(ABb) | PMID:26701906 | RRID:ZDB-GENO-960809-7 | Parichy Lab derivative of AB, AB$^{wp}$ |
| Strain, strain background (*Danio rerio*) | nkt; nackt$^{dt1261}$ | Gift.PMID:18833299 | RRID:ZDB-ALT-090324-1 | NA |
| Strain, strain background (*Danio rerio*) | hag; hagoromo | Gift.PMID:10801422 | RRID:ZDB-ALT-040217-6 | NA |
| Strain, strain background (*Danio rerio*) | sp7:EGFP; Tg(sp7:EGFP)$^{b1212}$ | Gift.PMID:20506187 | RRID:ZDB-ALT-100402-1 | NA |
| Strain, strain background (*Danio rerio*) | hs:dkk1; Tg(hsp70l:dkk1b-GFP)$^{w32T}$ | Gift.PMID:17185322 | RRID:ZDB-ALT-131120-19 | NA |
| Strain, strain background (*Danio rerio*) | 7xTCF:mCherry; Tg(7xTCF-Xla.Siam:nlsmCherry)$^{ia5}$ | Gift.PMID:22546689 | RRID:ZDB-TGCONSTRCT -110113-2 | NA |
| Recombinant DNA reagent | *Eda-2A-mCherry*; hsp70l:eda-2A-nls-mCherry | This paper | NA | Assembled using multisite gateway cloning |
| Recombinant DNA reagent | *Edar-2A-mCherry*; hsp70l:edar-2A-nls-mCherry | This paper | NA | Assembled using multisite gateway cloning |
| Antibody | anti-Dig-AP, Fab fragments | Millipore-Sigma | Roche Cat# 11093274910, RRID:AB_514497 | 1:5000 overnight at 4°C |

*Continued on next page*

*Continued*

| Reagent type (species) or resource | Designation | Source or reference | Identifiers | Additional information |
|---|---|---|---|---|
| Antibody | rabbit anti-beta-catenin antibody | Millipore-Sigma | Millipore cat#PLA0230, RRID:AB_2732045 | 1:1000 overnight at 4°C |
| Chemical compound, drug | Alizarin-Red-S; ARS | Millipore-Sigma | SKU_A5533 SIGMA-ALDRICH | NA |
| Chemical compound, drug | Calcein | Millipore-Sigma | SKU_C0875 SIGMA | NA |
| Chemical compound, drug | BGJ398 | Selleckchem | CatalogNo_Selleckchem: S2183 | NA |
| Chemical compound, drug | cyclopamine | Selleckchem | CatalogNo_Selleckchem: S1146 | NA |
| Software, algorithm | GraphPad Prism | GraphPad | NA | NA |

## Fish

Fish were maintained in the WT(ABb) background at 28.5°C. Lines used were $Tg(sp7:EGFP)^{b1212}$ abbreviated *sp7:EGFP* (*DeLaurier et al., 2010*); $Tg(hsp70l:dkk1b-GFP)^{w32T}$ abbreviated *hs:dkk1*(*Stoick-Cooper et al., 2007*); $Tg(7xTCF-Xla.Siam:nlsmCherry)^{ia5}$ abbreviated *7xTCF:mcherry* (*Moro et al., 2012*). $Tg(Mmu.Foxa2-cryaa:mCherry)^{nu15Tg}$ abbreviated *gli2-D:mCherry* (*Schwend et al., 2010*); $Tg(hsp70l:gli2aDR-EGFP)^{umz33Tg}$ abbreviated *hs:gli2-DR* (*Shen et al., 2013*); $Tg(hsp70l:shha-EGFP)^{umz30Tg}$ abbreviated *hs:shha* (*Shen et al., 2013*); $Tg(-8.0cldnb:LY-EGFP)^{zf106Tg}$ abbreviated *cldnb:EGFP* (*Haas and Gilmour, 2006*); $nackt^{dt1261}$ abbreviated *nkt* (*Harris et al., 2008*); *hagoromo* abbreviated *hag* (*Kawakami et al., 2000*). For regeneration experiments, scales were removed using forceps.

## Molecular biology

All coding sequences and in-situ probe templates were amplified using Primestar-GXL (Takara) from cDNA prepared with SSIII (ThermoFisher) and cloned into pJet1.2/blunt (ThermoFisher) with the following primers: *lef1* 5'tgtagggtgaggaggactttca, 5'cctgtagctgctgtctttgctt; *axin2* 5'agggataatat-taagcgtcagcag, 5'ggccctttttgaagaagtatctgta; *eda* 5'agaggacgaggaagttcggtat; 5'gtgcatgtgttcaggtttggta; *edar* 5'ttacggcactaaagacgatgatta; 5'ggattagtgcagttctgtgttcc; *fgfr1a* 5'tcagaaagtgctgatgtcctagtc, 5'cataagtctgcacacacacacact, *fgfr2* 5'aattcgctgtctgctctttttct, 5'gtctcagtgtttttgagaactgga; *shha* 5'acaacgagaaaccctgctagac; 5'gtctctctctcactctcgctctct; *hhip* 5'tcag-cagtcctgtttatttctgag, 5'gtaacattgccaaatggtgaagag. *hsp70l:eda-2A-nls-mCherry* abbreviated *Eda-2A-mCherry,* and *hsp70l:edar-2A-nls-mCherry* abbreviated *Eda-2A-mCherry*, were made using the *tol2* Gateway Kit (*Kwan et al., 2007*), and injected together with *tol2* mRNA (*Kawakami, 2004*).

## In-situ hybridization

In-situ probes and tissue were prepared as described previously (*Quigley et al., 2004*). Probes were hybridized for 24 hr at 66°C. Post-hybridization washes were performed using a BioLane HTI 16Vx (Intavis Bioanalytical Instruments), with the following parameters: 2x SSCT 3 × 5 min, 11 × 10 min at 66°C; 0.2x SSCT 10 × 10 min; blocking solution [5% normal goat serum (Invitrogen), 2 mg/mL BSA (RPI) in PBST] for 24 hr at 4°C; anti-Dig-AP, Fab fragments (1:5000 in blocking solution, Millipore-Sigma) for 24 hr at 4°C; PBST 59 × 20 min. AP staining was performed as described (*Quigley et al., 2004*). Tissue for sectioning was equilibrated into 5% gelatin (300-bloom, type-A, Sigma), post-fixed in 4% PFA/PBS overnight at 4°C and sectioned on a Vibratome 1500 (Harvard Apparatus).

## Immunostaining

Tissue was fixed in freshly prepared 4% PFA/PBS (EMS) for 1 hr at 4°C, blocked using blocking solution (described above) for 24 hr at 4°C and incubated with rabbit anti-beta-catenin antibody (1:1000 in blocking solution, Sigma PLA0230) for 24 hr at 4°C. Secondary antibody (Alexa Fluor 568 Goat anti-rabbit; Life-Technologies ALL036) was applied at 1:400 in blocking solution. 3 μM DAPI and 130

nM Alexa Fluor-488 Phalloidin (ThermoFisher) were used as counterstains. 12 × 20 min PBST washes were performed following both primary and secondary antibodies.

### Heat-shock induction

All heatshocks were performed in a 10 gallon glass aquarium equipped with a 1000 watt submersible heater and a programmable temperature controller (Process Technology). Larvae were given 6 × 1 hr 39°C heat shocks per day. Early *hs:dkk1* and *hs:gli2-DR*: PR stage larvae (*Parichy et al., 2009*) were selected and heat-shocked for 10 d. Late *hs:dkk1*: 2R stage larvae (*Figure 1B*) were selected and heat-shocked for 3 d. *hs:Eda-2A-mCherry* and *hs:Edar-2A-mCherry*: PR stage larvae were selected and heat-shocked for 5 d. For scale regeneration, heatshock induction began 16 hr prior to scale removal and was maintained throughout regeneration timecourse.

### Drug treatments

BGJ398: 2R stage larvae (*Figure 1B*) were selected and incubated daily in 1 μM BGJ398 (Selleckchem)(*Guagnano et al., 2011*) or 0.1% DMSO for 1 hr over 5 d. Cyclopamine: PR stage larvae were selected and incubated daily in 40 μM cyclopamine (Selleckchem [*Gould and Missailidis, 2011*]) or 0.25% ethanol for 2 hr over 12 d. All larvae were housed individually in 100 mL system water and fed freshly-hatched Artemia (Aquacave) between treatments.

### Vital dyes

Vital dyes were dissolved in fish system water and titrated to pH 7.5 with sodium bicarbonate. Fish were incubated in 0.4% Alizarin Red-S (ARS) or 0.2% calcein (Sigma) for 1 hr at 28.5°C and washed for at least 1 hr in fresh system water.

### Imaging

Live imaging: larvae were immobilized using 0.2% Tricane Methanesulfonate (Western Chemical) and imaged on an inverted Zeiss AxioObserver microscope equipped with Yokogawa CSU-X1M5000 laser spinning disk. In-situ: Zeiss AxioObserver or AxioZoom V16 uprigt stereomicroscope. Immunostainings: Zeiss LSM-800 scanning confocal microscope. Brightness and contrast were adjusted using Adobe Photoshop when necessary.

### Quantification and statistics

Quantifications were made using ImageJ and analyzed using GraphPad Prism. All tests of statistical significance used two-tailed, unpaired t-tests with Welch's correction.

## Acknowledgements

Funded by NIH R35 GM122471 and University of Washington Royalty Research Fund #A112414 to DMP. We thank S Halabiya for assistance with molecular biology, T Piotrowski, R Dorsky, R Karlstrom, T Becker and M Harris for reagents, and Parichy Lab members for animal care.

## Additional information

### Funding

| Funder | Grant reference number | Author |
| --- | --- | --- |
| National Institutes of Health | R35 GM122471 | David M Parichy |
| University of Washington Royalty Research Fund | A112414 | David M Parichy |

The funders had no role in study design, data collection and interpretation, or the decision to submit the work for publication.

## Author contributions
Andrew J Aman, Conceptualization, Data curation, Formal analysis, Funding acquisition, Investigation, Methodology, Writing—original draft; Alexis N Fulbright, Investigation, Optimized drug incubation procedures. Performed, Documented and analyzed cyclopamine experiments; David M Parichy, Conceptualization, Resources, Data curation, Formal analysis, Supervision, Funding acquisition, Visualization, Methodology, Project administration, Writing—review and editing

## Author ORCIDs
Alexis N Fulbright  https://orcid.org/0000-0002-5634-4094
David M Parichy  http://orcid.org/0000-0003-2771-6095

## Ethics
Animal experimentation: This study was performed in strict accordance with the recommendations in the Guide for the Care and Use of Laboratory Animals of the National Institutes of Health. All of the animals were handled according to approved Animal Care and Use Committee (ACUC) protocols (#4170) of the University of Virginia. All imaging and regeneration experiments were performed under MS222 anesthesi, and every effort was made to minimize suffering.

## Decision letter and Author response
Decision letter https://doi.org/10.7554/eLife.37001.019
Author response https://doi.org/10.7554/eLife.37001.020

# Additional files

## Data availability
All data generated or analyzed during this study are included in the manuscript and supporting files. Specialized materials including transgenic and mutant lines are available upon request.

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
