## [Decision Letter]

Thank you for submitting your article "Wnt/β-catenin regulates an ancient signaling network during zebrafish scale development" for consideration by *eLife*. Your article has been reviewed by three peer reviewers, one of whom is a member of our Board of Reviewing Editors and the evaluation has been overseen Didier Stainier as the Senior Editor. The reviewers have opted to remain anonymous.

In this landmark study by Aman et al., the authors use the zebrafish to analyze the signaling requirements for scale development. They make good use of mutant and transgenic lines to both image and perturb this process. They reveal a surprising degree of evolutionary conservation in zebrafish scale development with that of other appendages such as chicken. These data are convincing, and also open up the zebrafish to further downstream studies using the system. Also, by providing a staging convention for squamation, similar to what they have done for pigment cells, this paper will be of general use to other investigators studying appendage patterning.

Overall, this is a very well done study and we are happy to move forward with acceptance once a few textual edits are addressed. The main issue is that some of the evolutionary connections were thought to be too strongly worded, and it would be helpful if you could modify the tone to address the comments raised by reviewer #3 below. We are not requesting any additional experiments.

Reviewer #1:

In this study by Aman et al., the authors use the zebrafish to analyze the signaling requirements for scale development. They make good use of mutant and transgenic lines to both image and perturb this process. They reveal a surprising degree of evolutionary conservation in zebrafish scale development with that of other appendages such as chicken. These data are convincing, and also open up the zebrafish to further downstream studies using the system. Also, by providing a staging convention for squamation, similar to what they have done for pigment cells, this paper will be of general use to other investigators studying appendage patterning. I have only a few minor questions that can be addressed in the text:

1) What is known about the relationship of the *sp7* transcription factor and regulation of the downstream signaling nodes (i.e. Wnt, HH) they have identified. Is *sp7* in the fish known to regulate or converge with these in turning on downstream genes? Are there likely other cell-type specific transcription factors that might act as master regulators?

2) Is there a relationship between scale emergence and any of the known pigment cells that are similarly emerging during this time point of development, and are these functionally connected to each other?

Reviewer #2:

The manuscript by Aman et al., addresses the mechanisms of formation of zebrafish scales. This represents an interesting system, as it provides a good model for studying the formation of a simple bone structure, as well as having the potential to reveal evolutionary insights by comparing the mechanisms of formation of scales across different species. This paper focuses on the signaling pathways that might control scale regeneration. Through the combination of imaging of transgenic lines and pharmacological and genetic perturbations, the authors demonstrate that osteoblasts follow a precise developmental program of differentiation to generate scales. This process is driven by the interplay of several signaling pathways. Wnt is the most upstream regulator, activating the Edar/Eda, Shh and Fgf pathways, which have different effects on scale induction. Overall, the experiments are well executed, and the results properly interpreted. I find the findings important and that they represent a significant extension over a similar recent paper by Iwasaki et al., 2018. For these reasons, I would recommend publication of the paper.

Reviewer #3:

This is a landmark paper on the development of zebrafish scales. The images are stunning and provide new insight into how the scales initiate and pattern in the adult fish. The authors use genetics and imaging to identify the key signaling pathways that regulate these processes. Overall, the work is very well done, and provides a new staging system for the scale developmental process. The work will be of interest to a broad audience, in part because of the imaging that captures these processes for the first time.

1) While the experimental work is well done, the interpretation within evolution is less convincing. They often frame their work within the context of addressing how amniote skin appendages and teleost scales arise from a common ancestral organ. While this is a very interesting idea, and there is evidence for this, it is not clear how this specific study contributes to this idea beyond what is already known. The authors need to explain this more clearly, or tone down their claims.

Examples where these statements could be rewritten include:

"Our observations of scale development indicate that sequential addition of skin appendage primordia, leading to a tightly packed hexagonal grid, is a feature of epidermal appendage development common to both amniotes and anamniote fishes."

"Taken together, our analyses of teleost scale developmental genetics using zebrafish suggest a single origin of skin patterning mechanisms in ancient fishes that has been conserved in extant vertebrates, even as the final adult morphology of feathers, hairs and scales appear wildly divergent."

"These analyses produce clear evidence for a single, ancient origin of vertebrate skin appendages, reflected in common developmental genetics of widely divergent appendage types across phylogenetically disparate vertebrate lineages."

These seem like overstatements, and it is not clear to me how the authors can state this based on their experiments.

2) The signaling pathways identified are common for many developmental processes, and it is their downstream targets and tissue specific control that will most likely determine how they contribute to scale formation. The conceptual conclusion from the work presented here also seems to step too far into the evolution model. For example: "Since HH signaling is also necessary for epidermal morphogenesis during hair and feather development (McKinnell et al., 2004; St. Jacques et al., 1998), the functional outputs of signaling on morphogenesis are also likely conserved and anciently evolved. In turn, this suggests a previously unappreciated role for invasive migration in epidermal morphogenesis that may be of general relevance to understanding cellular mechanisms underlying skin appendage development." These are rapid conceptual jumps based on the evidence presented here on pathways widely used in many developmental systems and would benefit from rephrasing.

3) The authors begin by following the transgenic expression of *sp7:EGFP* which is said to mark osteoblasts. The authors need to explain the *sp7:EGFP* transgenic line, and the specificity of the *sp7* promoter.

4) Some of the text is subjective and should be toned down.

For example, the word unprecedented in this sentence: "Our analysis, based on live imaging of individual fish, has revealed the developmental anatomy of squamation and individual scale development with unprecedented resolution. "

Further, in the Introduction the justification that it is an understudied area is also distracting, such as the word "inattention": "This inattention perhaps reflects perceived technical challenges to working with larval fish, as compared to embryos." Here, and throughout, the text would make more of an impact to keep it based on the science.

5) Are the *cldnb:EGFP*+ epidermis in *hs:shha* truly migrating off the fish onto the glass cover slip? Perhaps they are sluffling off the fish and easily adhering to the glass?

---

## [Author Response]

Reviewer #1:In this study by Aman et al., the authors use the zebrafish to analyze the signaling requirements for scale development. They make good use of mutant and transgenic lines to both image and perturb this process. They reveal a surprising degree of evolutionary conservation in zebrafish scale development with that of other appendages such as chicken. These data are convincing, and also open up the zebrafish to further downstream studies using the system. Also, by providing a staging convention for squamation, similar to what they have done for pigment cells, this paper will be of general use to other investigators studying appendage patterning. I have only a few minor questions that can be addressed in the text:1) What is known about the relationship of the sp7 transcription factor and regulation of the downstream signaling nodes (i.e. Wnt, HH) they have identified. Is sp7 in the fish known to regulate or converge with these in turning on downstream genes? Are there likely other cell-type specific transcription factors that might act as master regulators?

In mammals, *sp7* regulates expression of secreted Wnt inhibitors Dkk1 and Sost (Zhang, 2012). There are almost certainly cell-type specific transcription factors that regulate scale development. Elucidating roles of feedback inhibitors and cell type specific transcription factors was, unfortunately, outside the scope of the current manuscript but is the subject of ongoing investigation in our lab.

2) Is there a relationship between scale emergence and any of the known pigment cells that are similarly emerging during this time point of development, and are these functionally connected to each other?

There does not appear to be a functional relationship between hypodermal pigment cells and scales. Scaleless *nkt* mutants, for example, produce normal stripes (Harris et al., 2008). Rather, it is likely that both pigmentation and scale development rely on common global regulators of developmental progression such as thyroid hormone. This is an area of ongoing investigation in our lab.

Reviewer #3:[…] 1) While the experimental work is well done, the interpretation within evolution is less convincing. They often frame their work within the context of addressing how amniote skin appendages and teleost scales arise from a common ancestral organ. While this is a very interesting idea, and there is evidence for this, it is not clear how this specific study contributes to this idea beyond what is already known. The authors need to explain this more clearly, or tone down their claims.Examples where these statements could be rewritten include:"Our observations of scale development indicate that sequential addition of skin appendage primordia, leading to a tightly packed hexagonal grid, is a feature of epidermal appendage development common to both amniotes and anamniote fishes."

This is not intended as an evolutionary hypothesis but a statement of empirical fact. Sequential addition of appendage primordia occurs in both fish and chickens (Chuong et al., 2013), and we have shown it to occur in zebrafish as well.

"Taken together, our analyses of teleost scale developmental genetics using zebrafish suggest a single origin of skin patterning mechanisms in ancient fishes that has been conserved in extant vertebrates, even as the final adult morphology of feathers, hairs and scales appear wildly divergent."

We rephrased to:

“Taken together, our analyses of teleost scale developmental genetics using zebrafish are consistent with a single origin of skin patterning mechanisms in ancient fishes that has been conserved in extant vertebrates, even as the final adult morphology of feathers, hairs and scales appear wildly divergent.” (Discussion section).

"These analyses produce clear evidence for a single, ancient origin of vertebrate skin appendages, reflected in common developmental genetics of widely divergent appendage types across phylogenetically disparate vertebrate lineages."

We rephrased to:

“These analyses show that scale development relies on signaling interactions similar to interactions that regulate the patterning and morphogenesis of amniote skin appendages such as hair and feathers and support a model in which diverse skin appendages of vertebrates arose from a common archetype.” (Introduction).

These seem like overstatements, and it is not clear to me how the authors can state this based on their experiments.

Our inferences are based on the observations that (i) specific requirements for signaling activity are similar between fish scales and amniote skin appendages, and (ii) the connectivity of these pathways is also similar. Since signaling pathways can wire together differently in different developmental contexts (Aman and Piotrowski, 2008; Arkell and Tam, 2012; Armstrong and Bischoff, 2004; Genikhovich et al., 2015; Monuki, 2007; Piran et al., 2009), the most parsimonious explanation for common regulatory networks is common origin. Nevertheless, since it is impossible to rule out convergence of regulatory architecture, we have qualified our evolutionary interpretations of the experiments with the above edits.

2) The signaling pathways identified are common for many developmental processes, and it is their downstream targets and tissue specific control that will most likely determine how they contribute to scale formation. The conceptual conclusion from the work presented here also seems to step too far into the evolution model. For example: "Since HH signaling is also necessary for epidermal morphogenesis during hair and feather development (McKinnell et al., 2004; St-Jacques et al., 1998), the functional outputs of signaling on morphogenesis are also likely conserved and anciently evolved. In turn, this suggests a previously unappreciated role for invasive migration in epidermal morphogenesis that may be of general relevance to understanding cellular mechanisms underlying skin appendage development." These are rapid conceptual jumps based on the evidence presented here on pathways widely used in many developmental systems and would benefit from rephrasing.

We rephrased to:

“During late steps in scale development, we found that Wnt/β-catenin-Eda dependent HH signaling regulates epidermal morphogenesis. Since HH signaling is also necessary for epidermal morphogenesis during hair and feather development (McKinnell et al., 2004; St-Jacques et al., 1998), it is possible that functional outputs of skin appendage signaling networks are conserved and anciently evolved.” (Discussion section).

3) The authors begin by following the transgenic expression of sp7:EGFP which is said to mark osteoblasts. The authors need to explain the sp7:EGFP transgenic line, and the specificity of the sp7 promoter.

Text has been added to more thoroughly introduce the *sp7* transcription factor and the *sp7:EGFP* transgenic line. (subsection “*sp7*+ osteoblast-like cells generate scales and reveal amniote-like skin patterning in zebrafish”).

4) Some of the text is subjective and should be toned down.For example, the word unprecedented in this sentence: "Our analysis, based on live imaging of individual fish, has revealed the developmental anatomy of squamation and individual scale development with unprecedented resolution."

We rephrased to:

“Our analysis, based on live imaging of individual fish, has revealed the developmental anatomy of squamation and individual scale development.” (Discussion section).Further, in the Introduction the justification that it is an understudied area is also distracting, such as the word "inattention": "This inattention perhaps reflects perceived technical challenges to working with larval fish, as compared to embryos." Here, and throughout, the text would make more of an impact to keep it based on the science.

We removed our interpretation of why scales have not received more attention. (Introduction).

5) Are the cldnb:EGFP+ epidermis in hs:shha truly migrating off the fish onto the glass cover slip? Perhaps they are sluffling off the fish and easily adhering to the glass?

This is possible, but filopodial extensions on leading edge cells suggest migratory behavior. Nevertheless, we have modified this section to more precisely describe the result without invoking migration per se. (Results section).